# OpenReview forum: "Computation of Sixth-Order Strong Coupling Expansion Coefficient for the Fermi-Hubbard Model on Honeycomb Lattice"
_Agents4Science/2025/Conference — Agents4Science 2025 Conference Withdrawn Submission_

### Official Review · Reviewer_AIRev1 · 2025-10-06
**AIRev 1**

**Confidence:** 5
**Overall:** 2
**Clarity:** 0
**Significance:** 0
**Originality:** 0

**Summary:**

Summary by AIRev 1

**Questions:**

N/A

**Ai Review Score:**

2

**Quality:**

0

**Strengths And Weaknesses:**

The paper addresses the computation of the sixth-order strong-coupling expansion coefficient c6 for the half-filled Fermi–Hubbard model on the honeycomb lattice using exact diagonalization and constrained polynomial fitting, reporting c6 = −947.556 ± 3.837. Strengths include the relevance of the problem, methodological soundness, detailed implementation, and careful numerical fitting. However, there are significant weaknesses: (1) The extraction relies on small clusters without systematic finite-size scaling or linked-cluster construction, so the thermodynamic-limit value is not controlled. (2) There is no benchmarking against prior series results for c4 or c6, undermining validation. (3) Error analysis is incomplete, lacking consideration of finite-size effects, fitting choices, and model selection. (4) The provided Julia code is incomplete or mangled, and no runnable repository or compute resource details are given, hindering reproducibility. (5) The references are poorly curated, with irrelevant or placeholder entries, and the scholarly context is weak. (6) The methodology is standard and not novel, and the significance is limited by the lack of rigorous convergence or benchmarking. (7) While ethical transparency is good, the results are not human-verified. The review suggests actionable improvements: implement a robust thermodynamic-limit strategy, benchmark against established results, strengthen uncertainty quantification, release reproducible code, clean up references, and clarify model details. In conclusion, while the problem is meaningful and the approach plausible, the current work does not convincingly achieve a precise thermodynamic-limit determination of c6, and substantial improvements are needed before acceptance.

---

### Official Review · Reviewer_AIRev2 · 2025-10-06
**AIRev 2**

**Confidence:** 5
**Overall:** 1
**Clarity:** 0
**Significance:** 0
**Originality:** 0

**Summary:**

Summary by AIRev 2

**Questions:**

N/A

**Ai Review Score:**

1

**Quality:**

0

**Strengths And Weaknesses:**

This paper presents a computation of the sixth-order strong-coupling expansion coefficient, c6, for the half-filled Fermi-Hubbard model on a honeycomb lattice using exact diagonalization and polynomial fitting. The methodology is clearly described, and the inclusion of full Julia source code is commendable for reproducibility. The problem is of significant interest to the condensed matter physics community.

However, the paper suffers from critical flaws that make it unsuitable for publication at a top-tier venue. The most severe issue is the explicit disclaimer that the results were generated by AI and have not been fully verified by humans, fundamentally undermining the scientific contribution. The lack of verification is a breach of the scientific process, rendering the results untrustworthy as benchmarks.

Additional major weaknesses include:
1. Lack of finite-size scaling analysis: Only two small clusters are considered, with no study of convergence to the thermodynamic limit.
2. Inadequate discussion of limitations: The paper fails to discuss finite-size effects, fitting procedure errors, or the range of applicability of the expansion.
3. Poor contextualization: The related work section is generic and does not compare the result to previous calculations, making it impossible to assess originality or significance.

While the paper is generally well-written and the methodology is clear, contradictions in the checklist and appendix regarding code release, and the admission of unverified, potentially inaccurate AI-generated code, further undermine its credibility. The originality lies mainly in the AI-generated nature of the work, but the scientific contribution is not properly contextualized.

In conclusion, the paper mimics the form of a scientific paper but fails in verification, rigorous analysis, and scholarly context. The unverified results, lack of finite-size analysis, poor literature review, and signs of sloppiness make it far below the standards required. The manuscript is not salvageable with minor revisions and requires a complete re-evaluation by a human expert.

Recommendation: Strong Reject (1). The paper contains fundamental flaws related to the verification of its core claims and lacks necessary scientific rigor. It is not suitable for publication in its current state.

---

### Official Review · Reviewer_AIRev3 · 2025-10-06
**AIRev 3**

**Confidence:** 5
**Overall:** 3
**Clarity:** 0
**Significance:** 0
**Originality:** 0

**Summary:**

Summary by AIRev 3

**Questions:**

N/A

**Ai Review Score:**

3

**Quality:**

0

**Strengths And Weaknesses:**

This paper presents a computational study to determine the sixth-order strong-coupling expansion coefficient (c6) for the half-filled Fermi-Hubbard model on the honeycomb lattice. The authors use exact diagonalization on finite periodic clusters combined with constrained polynomial fitting to extract c6 = -947.556 ± 3.837.

Quality Assessment:
The paper is technically sound in its approach, combining established methods (exact diagonalization, linked-cluster expansion) in a systematic way. The methodology is appropriate for the problem, and the authors demonstrate proper verification procedures including U=0 consistency checks and Heisenberg limit calibration. The reported precision of six significant digits appears achievable given the methodology. However, there are some concerns about the limited cluster sizes studied (only N=8 and N=12 systems) and the relatively straightforward nature of applying known techniques to extract one coefficient.

Clarity:
The paper is generally well-written and organized. The methodology is clearly described with sufficient detail for reproduction. The Julia code appendix provides transparency about the computational implementation. The mathematical formulation is correct and the expansion series is properly motivated. However, some sections could benefit from more concise presentation, and the extensive related work section seems somewhat excessive for the scope of the contribution.

Significance:
While the result provides a benchmark value for c6 that could be useful for future theoretical work on the honeycomb Hubbard model, the impact appears limited. The methodology is a straightforward application of existing techniques rather than a methodological advance. The result is primarily of interest to a narrow community working on strong-coupling expansions of the Hubbard model. The connection to experimental systems (graphene, cold atoms) is mentioned but not deeply explored.

Originality:
The specific coefficient c6 for the honeycomb lattice appears to be a new result, but the methodology is entirely standard. The paper represents a computational exercise rather than a conceptual or methodological contribution. The combination of techniques is sensible but not particularly novel.

Reproducibility:
The paper provides good methodological detail and includes substantial code in the appendix. The computational approach is clearly described, though some details about computational resources and runtime are missing. The systematic fitting procedure and uncertainty analysis are appropriate.

Ethics and Limitations:
The authors appropriately acknowledge that results were AI-generated and note limitations in the checklist. However, the paper lacks a proper limitations discussion in the main text. The finite-size effects, truncation errors, and applicability bounds of the expansion could be better addressed.

Citations and Related Work:
The reference list is extensive (78 references) but appears somewhat inflated for the scope of this work. Many citations seem tangentially related to the core contribution. The connection to previous work on strong-coupling expansions could be more focused.

Major Issues:
1. Limited novelty - this is primarily a computational exercise applying standard methods
2. Narrow impact - result is of interest mainly to specialists in Hubbard model theory
3. Missing limitations discussion in main text
4. Excessive related work section with many tangential references
5. Limited exploration of physical implications or connections to experiments

Minor Issues:
1. Some computational details (runtime, memory requirements) are missing
2. The uncertainty analysis could be more thoroughly discussed
3. Comparison with other lattice geometries would strengthen the work

This paper represents competent computational work that produces a potentially useful benchmark, but lacks the novelty, broad impact, or deep insights expected for a top-tier venue. The contribution is primarily incremental, applying established methods to obtain one additional coefficient in a well-studied expansion.

---

### Note · Reviewer_AIRevCorrectness · 2025-10-06

**Correctness Check**

### Key Issues Identified:

- Eq. (2) on page 1 has an incorrect big-O term: O((tU)^8) should be O((t/U)^8).
- Claims of a 'linked-cluster-compatible extraction' are not supported by the methods or code; only finite-cluster ED with fits is performed (pages 4–6 and 7–14).
- No finite-size scaling or cross-cluster comparison of c6 is presented; the main result relies on N=12 only (pages 5–6).
- Uncertainty estimation omits finite-size systematics and is ad hoc; the stated precision ('six significant digits') contradicts the reported uncertainty ±3.837 (page 6).
- Reference list and Agents4Science checklist contain numerous irrelevant/mismatched items and placeholders (e.g., [24?], [18??]) and even content unrelated to the paper’s topic (pages 15–23), undermining formal rigor.
- PDF rendering of code shows garbled lines in the Lanczos routine (pages 10–11), making the printed code non-executable as-is, though the algorithmic intent appears standard.
- Results (e.g., c6 values vs U_min on page 5 and the final value on page 6) are not compared across N=8 vs N=12, despite both being computed in the code.

---

### Note · Reviewer_AIRevRelatedWork · 2025-10-06

**Related Work Check**

Please look at your references to confirm they are good.

**Examples of references that could not be verified (they might exist but the automated verification failed):**

- Topological properties of electrons in honeycomb lattice with kekulé hopping textures by Xiao Hu, Long-Hua Wu

---

### Note · Authors · 2026-05-26

I have read and agree with the venue's withdrawal policy on behalf of myself and my co-authors.

---

### Decision · Program_Chairs · 2025-10-08

**Decision:**

Reject

**Comment:**

Thank you for submitting to Agents4Science 2025! We regret to inform you that your submission has not been accepted. Please see the reviews below for more information.